# Polymeric Biomaterials for Topical Drug Delivery in the Oral Cavity: Advances on Devices and Manufacturing Technologies

**DOI:** 10.3390/pharmaceutics15010012

**Published:** 2022-12-20

**Authors:** Paula de Freitas Rosa Remiro, Mariana Harue Taniguchi Nagahara, Rafael Abboud Azoubel, Michelle Franz-Montan, Marcos Akira d’Ávila, Ângela Maria Moraes

**Affiliations:** 1Department of Engineering of Materials and of Bioprocesses, School of Chemical Engineering, University of Campinas, Campinas 13083-852, SP, Brazil; 2Department of Manufacturing and Materials Engineering, School of Mechanical Engineering, University of Campinas, Campinas 13083-860, SP, Brazil; 3Department of Biosciences, Piracicaba Dental School, University of Campinas, Piracicaba 13414-903, SP, Brazil

**Keywords:** oral cavity, drug delivery, polymers, biomaterials, topical administration

## Abstract

There are several routes of drug administration, and each one has advantages and limitations. In the case of the topical application in the oral cavity, comprising the buccal, sublingual, palatal, and gingival regions, the advantage is that it is painless, non-invasive, allows easy application of the formulation, and it is capable of avoiding the need of drug swallowing by the patient, a matter of relevance for children and the elderly. Another advantage is the high permeability of the oral mucosa, which may deliver very high amounts of medication rapidly to the bloodstream without significant damage to the stomach. This route also allows the local treatment of lesions that affect the oral cavity, as an alternative to systemic approaches involving injection-based methods and oral medications that require drug swallowing. Thus, this drug delivery route has been arousing great interest in the pharmaceutical industry. This review aims to condense information on the types of biomaterials and polymers used for this functionality, as well as on production methods and market perspectives of this topical drug delivery route.

## 1. Introduction

The most common route for drug delivery is the oral route [1,2], in which the drugs (tablets, capsules, syrup, solutions, suspensions, powder, emulsions, etc.) are placed in the mouth and swallowed. Then, the drug can be absorbed in the gastrointestinal tract. However, drug absorption can be affected by limited drug chemical and biological stability or by physiological barriers [2]. Another limiting factor for the use of the oral route of administration is that many people have difficulties in swallowing, especially the elderly and children [3,4]. Liquid and semi-solid formulations do not exhibit several of the problems concerning swallowing, but they often present palatability issues and can cause nausea and gastric discomfort. Added to these problems, the difficulty of local treatment of diseases affecting oral mucosa is another concern [5].

An alternative to the oral route of administration is transbuccal drug delivery, with advantages, such as bypassing the first-pass hepatic metabolism and avoiding drug degradation by gastrointestinal enzymes [6]. Moreover, topical administration is convenient and easy to access, minimally invasive, and presents higher drug bioavailability in comparison to transdermal administration [7]. Thus, the development of biomaterials capable of delivering drugs topically via the oral cavity is an attractive alternative for enhancing medication bioavailability and for drug administration in emergency situations, when intravenous application is impaired. However, despite the higher permeability of oral mucosa when compared to the skin, the epithelium consists in an efficient barrier to drug permeation and protects underlying connective tissues from damage [8,9,10,11,12]. Therefore, efficient transbuccal delivery remains a great challenge.

There is a range of biomaterials that can be used to deliver the drug through the oral mucosa. The drugs may or may not have systemic action and this depends on the type of device, dosage, and mucosa interaction, among other factors [13]. The biomaterials that are most widespread in the literature for this purpose are pressure tablets, fast-dissolving films and gels (oro-dissolving), mucoadhesive films and gels, and microneedles [7,14].

Biomaterials for drug delivery can be manufactured from numerous classes of materials. Nonetheless, polymers have very interesting characteristics for this purpose, and a large range of flexible, versatile, and non-toxic molecules can be selected among them for transbuccal delivery. These polymers should have a particular set of characteristics [15,16]. These characteristics include being non-toxic, non-irritating, and free of leachable substances. The polymers should not promote irritation and infection of the oral mucosa. In addition, the mechanical properties of the polymers should be compatible with the type of application, and the films formed by these polymers should have no taste. If they do, they should be palatable. Finally, the polymers should be easily accessible, of low cost, and should show adequate shelf-life and proper mucoadhesive properties.

Polymers can be roughly divided into synthetic and natural classes. The first includes those produced in chemical industries and the second, those that can be extracted from a natural source. It is also possible to have natural polymers chemically modified to have improved properties, and another subclass, called smart polymers, which represents molecules capable of responding to changes in the surrounding microenvironment [17].

Polymeric biomaterials containing drugs can be prepared in several ways, e.g., by solvent casting, direct compression, hot melting extrusion, rolling method, and 3D printing [18,19]. The best technique will depend on the nature of the polymer, the required architecture of the biomaterial, its application site, and the dosage of drug required for the treatment.

Figure 1 shows the increasing interest in polymeric oral cavity devices lately. As a result, it is reasonable to foresee that, shortly, polymeric devices will play an essential role in oral healthcare. Since polymeric formulations for drug delivery via topical application in the oral cavity represent a very promising and interesting therapeutic approach, this review addresses the types of polymeric biomaterials used for this purpose, their production methods, and the market perspectives related to this drug delivery route.

## 2. Types of Polymeric Biomaterials Commonly Used in Drug Delivery in the Oral Cavity

The main types of biomaterials used for drug delivery in the oral cavity are fast-dissolving films and gels, mucoadhesive tablets, films and gels, and microneedles, which will be discussed below.

### 2.1. Fast-Dissolving Films (Oro-Dissolving)

An oral fast-dissolving film consists of a very thin oral strip that is simply placed on the patient’s tongue or any oral mucosal tissue. The film, instantly wet by saliva, rapidly hydrates and adheres to the site of application, where it disintegrates and dissolves, releasing the medication [20,21]. Oral dissolving films have large surface areas, which facilitates their fast disintegration in the oral cavity. The drug can then be absorbed directly and reach the systemic circulation. These biomaterials do not require special storage or transport conditions as they are flexible, compact, and have a long shelf-life. These formulations can be used directly by the patient, anywhere, and at any time. They can be produced with less drug quantities, as the first-pass loss through the hepatic metabolism effect is negligible, which reduces the drug’s side effects. These dosage forms are also favorable for those suffering from dysphagia, repeated vomiting, hypertension, heart attack, asthma, nausea, paralysis, and mental disorders [15,22,23]. This type of biomaterial also allows administering high drug concentrations faster when compared to conventional tablets, for example.

There is a wide variety of polymers that can be useful for producing fast-dissolving films. The use of these materials has aroused great interest in the medical and nutraceutical fields. The polymers can be used alone or combined with other materials, depending on the required attributes of the film. For this application, the polymers should be water-soluble, as they must dissolve in saliva, showing a high wettability rate, but appropriate disintegration time [15,16]. Some of the polymers and drugs used to produce oral fast-dissolving films are listed in Table 1.

Figure 2 illustrates a representative diagram of an example of one of the fabrication techniques of an oral dissolving film, up to the moment of its application.

If fast film dissolution and rapid drug release are not desired, an alternative to prolonged drug release can be the use of mucoadhesive biomaterials, as discussed below.

### 2.2. Mucoadhesive Biomaterials

Mucoadhesion can be defined as the attachment of macromolecules to mucous membranes. Mucoadhesive formulations are useful since products without these characteristics can be leached and ingested by patients, causing interruptions or reductions in drug absorption during the treatment [37,38,39]. Adhesion of a material to a mucous membrane or a mucus-covered surface has been employed to prolong drug contact with adsorption sites and, consequently, to improve drug absorption [40].

The mechanisms involved in mucoadhesion are not yet fully elucidated, with divergence in the methods used to quantify the interaction between polymers and mucosa [41]. Numerous polymers can be used in the oral region to achieve mucoadhesion. These polymers should have the ability to hydrate and swell when in contact with the mucus-lined epithelium [37,42]. The mucoadhesive ability of a dosage form is dependent upon a variety of factors, including the nature of the mucosal tissue and the physicochemical properties of the polymeric formulation [43]. The most common mucoadhesive biomaterials are described below.

#### 2.2.1. Buccal Tablets

Buccal tablets are easy to prepare, as they can be made only by compression, a process easy to scale up, efficient, and an economic method for large-scale production. However, among the formulations generally used for oral purposes, it is considered the least comfortable. Despite this limitation, mucoadhesive buccal tablets can be loaded with a greater amount of drug than films and gels, being useful for treatments in which the required drug concentration is high, and slow drug release is desired in comparison to fast-dissolving films [37,39,43]. An example of a mucoadhesive buccal tablet and a summarized mechanism of how it remains adhered to the mucosa are shown in Figure 3.

Numerous works in the literature report the use of buccal tablets for drug delivery. Koirala et al. (2021) evaluated tablets of aceclofenac for sustained drug release, to improve patient compliance for the management of different types of pain [44]. Çelik et al. (2017) developed a study to design and optimize risperidone mucoadhesive buccal tablets for systemic delivery routes and concluded that this biomaterial can be used as an alternative treatment for schizophrenia [37]. Abruzzo et al. (2015) prepared mucoadhesive tablets consisting of chitosan/gelatin microparticles compressed with propranolol hydrochloride for buccal delivery and concluded that this formulation can be effective for the treatment of hypertension, angina, atrial fibrillation, postinfarction, sinus tachycardia, arrhythmias, and obstructive cardiomyopathies [39]. Due to the route of administration, this type of biomaterial also allows bypassing the extensive hepatic first-pass, avoiding some side effects. Chandira et al. (2009) developed mucoadhesive tablets of clarithromycin, a macrolide antibiotic, which were designed to extend the gastric residence time after oral administration. The authors were able to obtain formulations capable of releasing more than 90% of the drug in 12 h, while keeping the physicochemical properties unchanged [45].

In another example, buccal adhesive tablets containing theophylline (a methylxanthine drug used in the therapy of respiratory diseases such as chronic obstructive pulmonary disease, asthma, or emphysema) were developed using direct compression [43]. The formulations were developed using a water-soluble resin with various combinations of mucoadhesive polymers. The theophylline tablets were evaluated for tensile strength, swelling capacity, and ex vivo mucoadhesion performance. The authors concluded that, in general, the majority of the developed formulations presented suitable adhesion and controlled drug release.

#### 2.2.2. Mucoadhesive Films and Gels

Among the formulations commonly used for transbuccal drug delivery (tablets, films, gels, pastes, and sprays), mucoadhesive films offer many advantages, such as high flexibility and large surface area, which implies a greater area for drug absorption [46,47,48]. Another relevant advantage is the fact that they ensure a higher drug dosage when compared to gels and other formulations such as pastes and sprays, since they can be leached by saliva [46,48]. They also have the advantage of being comfortable, as they are thin, flexible, and most of the time have mechanical properties suitable for their purpose.

Gels, in turn, have the limitation of not supplying the amount of drug to the mucosa in a homogeneous way, but they have some advantages over other types of formulation, such as the relatively faster release of the incorporated drug and easy preparation. In addition, their simple administration and greater mucoadhesiveness, allowing adhesion to the mucosa in the gingival pocket, as well as rapid elimination through normal catabolic pathways, which reduces allergic reactions at the application site [49], are attractive characteristics.

The use of mucoadhesive films and gels for drug delivery in the oral cavity is well consolidated in the literature. These films are being proposed for the adjuvant treatment of oral carcinomas [5,50], chronic pain [51], for the therapy of small buccal lesions [49], to treat symptoms of migraine headaches, for pressure control [46,52,53], to treat recurrent aphthous stomatitis [54], as a topical anesthetic in dentistry [55,56], and as an anxiolytic [57], among other applications.

### 2.3. Microneedles (MNs)

Microneedles are three-dimensional microstructures with microscale length (usually less than 1500 μm) that can break the barrier of transdermal drug delivery. They can pierce the stratum corneum and generate transient microchannels through which external molecules can passively diffuse into the skin [58]. These biomaterials can penetrate and release drugs into the skin, which is about 10–100 times less permeable than the mucosa [59]. These biomaterials have been employed for drug delivery, vaccination, bio-sensing, and diagnostic purposes, and analysis or current literature indicates that the development of microneedles tends to grow in the near future. For instance, microneedles have been shown to have potentially good effects as transmucosal delivery systems, as recently suggested in a randomized clinical trial [60].

Microneedles can access adequate tissue depths, being able to deliver the drug deep into tissue without stimulating nerves in the underlying tissue and damaging blood vessels. Thus, the treatment with microneedles enables a minimally invasive delivery of several molecules to the tissue (skin, mucosa) in a way that overcomes the limitations of conventional methods of transdermal drug administration [58]. Another advantage of these biomaterials is their versatility, as they can be used to deliver drugs not only to the oral cavity [61], but also to the skin [62,63] and eyes [64,65].

For the reasons discussed above, this type of biomaterial will be discussed in greater detail. Microneedles can be broadly classified based on their overall shape and tip shape, which are important design and fabrication issues. Different designs of microneedles have been proposed and fabricated, such as cylindrical, conical, pyramidal, candle, spike, spear, square, pentagonal, hexagonal, octagonal, and rocket shapes [66,67,68]. They can be produced using metals [69,70,71,72], glasses [73], silicon [74,75,76], polymers [77,78,79,80,81], ceramics [82], borosilicate [83], and carbon nanotube-polyimide [84], among other materials. Furthermore, different techniques of drug loading and delivery may be employed [66]. Figure 4 summarizes the microneedle types, fabrication materials, and methods.

#### 2.3.1. Types of Microneedles Concerning the Technique of Drug Loading and Delivery

Microneedles can be divided into five main classes: solid, coated, dissolving, hollow, and hydrogel-based microneedles, as illustrated in Figure 5, showing different working mechanisms.

Solid microneedles (SMNs) are used to mechanically disrupt the tissues and create transient pores of micron dimensions before administration of the active pharmaceutical ingredients (APIs) from the external reservoir. Therefore, in SMNs-based drug delivery, the SMNs are first pressed against the tissue surface, followed by the application of traditional patches, or other types of dosage forms, such as the ones mentioned before, containing drug molecules. This allows for molecules of different physico-chemical properties to be delivered by using SMNs [86,87]. Solid microneedles are easy to manufacture, and usually have better mechanical properties and sharper tips when compared to hollow microneedles [88].

Coated microneedles are those in which the drug coats the microneedle surface as a solid film. When the coated microneedle is inserted into the tissue, it carries the film containing the active ingredient. Once in the tissue, within a few minutes, the coating dissolves, releasing the drug, and then it is possible to remove the microneedles and safely discard them [89].

The coated microneedle is a multifunctional system and can be used to deliver numerous molecules, such as proteins and DNA, as well as viruses [90]. One of the advantages of coated microneedles is that their mechanical properties do not vary, as occurs with soluble microparticles. The structural base of these microneedles is fairly resistant, and only the coating material and drug go through changes [89].

Dissolving microneedles (DMs) are ultra-small needles composed of water-soluble materials, usually with lengths in the micrometer range (less than 1000 µm). They create pores in the tissue and release the active principle upon microneedles dissolution [91]. These biomaterials have several advantages, such as having no risk of leaving harmful materials in the tissue, and no generation of sharp needle waste. These biomaterials present a relatively low cost, are easy to produce, and their fabrication at industrial scale is feasible [92].

Dissolving microneedles are reported in the literature for several uses, such as transdermal delivery of huperzine A for the treatment of Alzheimer’s disease [93]. They can also be used for the delivery of ibuprofen, frequently used as a low molecular weight model drug demanding high dosages [94], and for the delivery of human growth hormones [95].

In fact, when it comes to delivery to the oral mucosa, soluble microneedles show a great rise lately in terms of the number of studies and publications. They can be configured as a new approach to drug delivery to the oral mucosa, effective and painless. Seon-Woo and co-workers developed a dissolving microneedle system for the oral mucosal to deliver triamcinolone acetonide to treat aphthous stomatitis [96].

Caffarell-Salvador et al. (2021) [97] designed a highly drug-loaded microneedle patch to deliver macromolecules and applied it to the buccal area, which allows faster delivery than through the skin. They successfully delivered 1-mg payloads of human insulin and human growth hormone to the buccal cavity of swine within 30 s. These are just a few examples of how dissolving microneedles can be a very useful tool for faster and more effective drug delivery.

Hollow microneedles are needles with an inner conduit for administering drugs in the tissue. These microneedles are similar to hypodermic injections, which allow pressure-actuated flow of a liquid formulation. The pressure, and consequently, the flow rate, can be modulated for fast and high dosage drug injection (bolus), slow infusion, or time-varying delivery rates [98].

Hydrogel-forming microneedles (HFMs) represent the newest form of microneedles, consisting of swellable polymers (crosslinked hydrogels) showing different performances from the microneedles mentioned above. When inserted into the tissue, HFMs swell due to the hydrophilic nature of the hydrogels, a property useful for many applications in biomedicine [99,100].

There are three types of drug loading methods in the case of hydrogel microneedles. Some microneedles only have drugs at the tips, some in patches, others have drugs in both places: needles and patches. One of the advantages of hydrogel microneedles is that the amount of drug loaded into the needles is greater than the possible amount to be loaded in solid and hollow microneedles. Another advantage of this type of microneedle is that it can be made from different types of polymers, making it possible to adjust the drug dosage according to the characteristics of the polymers used in its production. In fact, polymers and polysaccharides are very attractive for this purpose due to their high compatibility, degradability, and non-toxicity [101,102,103].

Zhu et al. (2022) developed a lidocaine-loaded hyaluronic acid adhesive microneedle patch for oral mucosal topical anesthesia. In this study, the authors reached the requirements for oral clinical application, i.e., rapid administration, water resistance, and adhesion, circumventing the disadvantages of the pain inflicted by direct local injection of anesthetic drugs and its ingestion, as well as the unpleasant taste of local surface anesthesia ointments [104].

Ye and coworkers (2016) developed a crosslinked hyaluronic acid microneedle patch coupled with pancreatic cells and enzymes for the delivery of insulin to regulate glucose levels, without the need for implantation, overcoming problems with the immune response and long-term efficacy of pancreatic cells therapy [105].

#### 2.3.2. Polymeric Microneedles

As mentioned above, microneedles can be manufactured with numerous materials, such as metals, ceramics, silicon, silica glass and carbohydrates. However, in this work the focus will be on discussing polymeric microneedles.

Polymeric microneedle (PMN) systems are interesting biomaterials because they can control drug delivery, have tunable properties, and are easy and practical for patient self-administration. They have the main advantages of being biocompatible, and easily and painlessly penetrate the stratum corneum or mucosa, delivering their contents into where they can be absorbed into the systemic circulation. These biomaterials allow controlling drug release kinetics. Depending on the application site, they reach specific tissues, as well as respond to changes in the physiological conditions of the surrounding environment [106]. Furthermore, polymeric microneedles are able to deliver a greater quantity of low molecular weight molecules for biological therapies and vaccines.

In addition to the conventional requirements of biocompatibility and biodegradability, solubility and mechanical properties are relevant regarding the production of polymeric microneedles. Polymeric microneedles are produced through three main ways. The first approach refers to coating the metal microneedles with the polymer and drug to be released, giving rise to coated polymeric microneedles. The second approach, the dissolving microneedles, requires the incorporation of the drug into the matrix of soluble polymeric microneedles. In this method, delivery efficiency is determined by the rate of polymer dissolution after insertion. The third approach allows drug delivery through passive diffusion, as in hydrogel microneedles, or polymer matrix degradation. This approach can be associated with a secondary encapsulation procedure (micro or nano-formulations) which can be adjusted in a bio-responsive way [107].

Dissolvable polymeric microneedles are considered the most effective approach for drug delivery and can be used in numerous applications. The drugs to be delivered can be incorporated into dissolvable and degradable polymers [107]. One of the benefits of this type of microneedle is the fact that the biomaterial itself carries the drug, and it is not necessary to apply it at two different times, as with solid microneedles. The major challenge of polymeric microneedles is their penetration into tissues. Polymers generally have lower mechanical strength than silicon and metals, and the penetration of these microneedles into the tissue can be impaired [108,109]. Therefore, two or more polymers or additional materials can be combined to increase the mechanical strength of MNs [109].

Frequently reported polymers for the production of dissolving MNs are poly(vinylalcohol) (PVA) [110,111,112], hyaluronic acid (HAc) [113,114], hydroxypropyl methylcellulose (HPMC) [115,116], carboxymethyl cellulose (CMC) [115,116], fluorescein isothiocyanate (FITC)-dextran [117,118], sodium alginate [119,120], and other biodegradable polymers, such as chitosan [119,121], polylactic acid (PLA) [122], and polyglycolic acid (PLGA) [123], among others.

There are several ways to produce polymeric microneedles, but the most common and scalable is using molding techniques. However, this technique has some limitations as it usually involves several laborious steps, such as preparation of the polymer formulation, mold design, and fabrication and plasticization of thermoplastic polymers, which may limit the use of thermosensitive drugs [124].

Another technique that has been widely used for the manufacture of polymeric microneedles is three-dimensional (3D) printing. This strategy gives the prototyping and manufacturing methods the flexibility to produce MN patches in a one-step manner with high levels of shape complexity and reproducibility [125]. This technique offers customization, cost-efficiency, a rapid response time between design iterations, and enhanced accessibility. The increase of printing resolution, the accuracy of the features, and the accessibility of low-cost raw printing materials have stimulated the use of 3D printing for the fabrication of microneedle platforms [126].

## 3. Steps Involved in the Production of an Oral Dispositive to Drug Delivery

Pires et al. (2015) [127], based on the work of Ratner et al. (2013) [128], outlined some important steps to identify the need for a new biomaterial, its manufacture, testing, regulation, sale, and deployment. In the present work, a parallel can be established for the development of any biomaterial for drug delivery in the oral cavity. Thus, the steps required from designing to placing a new formulation on the market for this purpose and its use by the patient are illustrated in Figure 6.

## 4. Polymers Used to Produce Biomaterials for the Use in Oral Cavity

The polymers commonly used for the production of biomaterials used in the oral cavity include natural, synthetic, and smart polymers, which are discussed in detail below.

### 4.1. Natural Polymers

According to the European Chemical Agency (ECHA) [129], natural polymers are those that result from a polymerization process that has taken place in nature, independently of the extraction process used in their production. Natural polymers have the advantage of having generally low toxicity, being biocompatible, biodegradable, and coming from renewable sources. However, this type of polymer can also present a high degree of variability from lot to lot, can be structurally more complex, and may have complex production, extraction, and purification processes of high cost [130].

Natural polymers can be classified in different types [131]. Polysaccharides of plant cell walls, for instance, originate from the cell wall of plants and include mainly cellulose, hemicelluloses, and pectin. Gums and mucilages are polysaccharides, converted to monosaccharides by hydrolysis [132], including guar gum, arabica gum, and karaya gum. Exudate gums are also produced by plants, but as a result of stress, e.g., physical injury or fungal attack. Some examples of this class include arabic gum, acacia gum, cashew gum, karaya gum, and tragacanth gum. Another class refers to inulin, which belongs to a class of carbohydrates known as fructans. The main sources of inulin used in the food industry are chicory and Jerusalem artichoke. However, some vegetables, such as garlic, asparagus, and Dahlia tubers, also have a considerable amount of inulin [133]. Within the plant-derived category, starch, a polysaccharide present in many green plants, may also be used, having a wide range of applications in the pharmaceutical sector as an excipient. There are also seaweed-derived polysaccharides, which are isolated from algae. Seaweed gums are represented by carrageenans, agar, and alginates.

Microbial polysaccharides correspond to molecules obtained as fermentation products of microorganisms such as *Xanthomonas campestris* bacteria (xanthan gum), *Pseudomonas elodea* (gellan gum), *Sclerotium* (scleroglucan), and the fungus *Aureobasidium pullulans* (pulullan). Dextrans, on the other hand, are glucose-based polymers with molecular weights ranging between 1000–40,000,000 Da, being produced by lactic acid bacteria and also by the dental plaque-forming species *Streptococcus mutans*. Dextran is a family of natural polysaccharides that is widely under investigation for use as polymeric carriers in novel drug delivery systems.

Finally, animal polysaccharides and proteins are obtained from animal sources. The most common polysaccharide extracted from animal sources is chitin, from which chitosan can be produced, while the main examples of this class of protein include collagen, gelatin, fibrinogen, silk, elastin, and keratin.

Table 2 presents the main natural polymers used to manufacture oral drug delivery biomaterials and examples of their applications.

### 4.2. Synthetic Polymers

Synthetic polymers are artificially produced through chemical reactions using well-defined conditions, e.g., heat, pressure, and catalyst type [139,140]. Most synthetic polymers have good mechanical properties, unlike natural polymers [140]. Compared to natural polymers, synthetic polymers are more biologically inert, the mechanical properties and degradation rates of synthetic polymers can be tailored, and, frequently, these polymers can be processed into various shapes. Since the synthesis conditions are well known and controlled, synthetic polymers offer higher batch-to-batch uniformity and show more reproducible physicochemical properties.

Synthetic polymers of interest in drug delivery for the oral cavity can be divided into different categories, e.g., polyethers, polyesters, poly (N-isopropylacrylamide-co-propylacrylic acid) copolymers, and poloxamers.

Among the polyethers, one of the most common is polyethylene glycol (PEG), which has gained great visibility due to its wide range of applications in the field of pharmacy. This compound is hydrophilic and can be used favorably as a penetration enhancer, especially in topical dermatological preparations. It is also widely used in cosmetics as cleansing surfactants, emulsifiers, skin conditioners, and humectants [141]. The two most important polyesters under intense investigation as drug carriers for active agent delivery are poly (lactic-co-glycolic acid) (PLGA), which is a copolymer of poly lactic acid (PLA) and poly glycolic acid (PGA), and polycaprolactone (PCL). Both can be biodegraded by hydrolysis of the ester linkages.

Poly (N-isopropylacrylamide-co-propylacrylic acid) copolymers (PNIPAAM) and their derivatives are also widely investigated for drug delivery purposes. They have great potential for the delivery of therapeutic proteins and peptides since they are thermoresponsive. However, the clinical use of PNIPAAM and its derivatives is limited, as they are non-biodegradable and, upon contact with blood, activate platelets.

Poloxamers, traded as Pluronic^®^ (BASF), Kolliphor^®^ (BASF), Lutrol^®^ (BASF), Synperonic^®^(Croda), and Antarox^®^ (Rhodia), are nonionic triblock copolymers. The central part of poloxamers is hydrophobic, composed of polypropylene oxide (PPO), and this region is flanked on both sides by polyethylene oxide (PEO) chains, which are hydrophilic. This class of polymers is thermosensitive and has been intensively studied for the sustained delivery of therapeutic proteins. These polymers are inert and known to maintain the stability of incorporated therapeutic proteins and peptides, with an increased in vivo half-life when compared to other drug delivery systems.

### 4.3. Smart Polymers

Smart polymers or stimuli-responsive polymers undergo reversible physico-chemical modifications with small alterations in their surrounding environment [142]. Frequently, these polymers exhibit a non-linear response to a small stimulus, leading to a macroscopic alteration in their structure or properties. Several changes can be observed in this sense, from swelling and contraction to complete disintegration. These variations can be caused by chemical events, including simple reactions, such as oxidation, acid-base reaction, reduction, and hydrolysis of fractions linked to the polymer chain. More severe changes can also occur, resulting in irreversible bond breakage in response to external stimuli, causing polymer structural degradation [143]. Physical-chemical changes can occur due to variation in pH, temperature, as well as due to the presence of enzymes and exposure to some types of radiation. Polymers that are stimulated via electric [144] and magnetic [145] fields are not as frequently reported in the literature for drug delivery, and therefore they will not be covered in this work.

#### 4.3.1. Temperature-Responsive Polymers

Thermoresponsive polymers are the most widely employed smart polymers, whose main characteristic is the reversible phase (or volume) transition that occurs in response to temperature changes. This property allows the manipulation of polymers in a remote and switchable way by controlling the temperature [30,146].

According to Hoffman (2013), different working mechanisms can be exploited for the development of temperature-responsive polymers. The three main classes of temperature-responsive polymers are: shape-memory materials, liquid crystalline materials, and responsive polymer solutions [146].

Shape-memory polymers (SMPs) belong to a class of smart materials that are mechanically modified by external stimuli. More directly, these materials can “remember” shapes. As a simple example, a complex three-dimensional SMP shape can be compressed into a smaller size shape (suitable for delivery of a catheter to the body or to fit in a compact space) by applying some source of heating or cooling, for instance. In a second moment, the application of heat, for example, can trigger a return to equilibrium, causing the object to go back to a complex shape through the mobilization of the network chain [147].

Liquid crystalline polymers (LCPs) show properties of both solids and liquids. These polymers can be of many types, depending on the position and type of the mesogenic units in the molecular architecture. LCPs are mainly classified as main-chain, side-chain, and crosslinked polymers, among others [148]. Main chain-type polymers elongate in the liquid crystalline phase and contract when heated in the isotropic phase, with full reversibility to the initial stage.

The most common types of thermoresponsive polymers are those that undergo a solution liquid-liquid phase transition in response to variation of the temperature, in which phase separation occurs from a homogeneous solution to a concentrated polymer phase and a dilute polymer phase. This effect is associated with the transition from a clear phase to a phase with clouds. Clouds are observed due to the formation of polymer-rich droplets, causing phase separation. When separation occurs at a high temperature, this is referred to as lower critical solution temperature (LCST) transition, while the reversed-phase behavior is known as upper critical solution temperature (UCST) transition [149].

The literature reports several examples of the use of temperature-responsive polymers for drug delivery in the oral cavity. Gao et al. (2010) developed a thermosensitive PLGA-PEG-PLGA hydrogel for the sustained release of docetaxel to treat lung cancer [150]. Choi et al. (2014) produced hydrogels with combined properties using the thermosensitive polymer Pluronic F127 (PF127) and the mucoadhesive polymer polyethylene oxide (PEO) to deliver the anticancer drug paclitaxel incorporated in dimethyl-β-cyclodextrin in the oral mucosa [151]. Shin et al. (2013) prepared a thermoreversible hydrogel of the polymer Pluronic F127 (PF127) with mucoadhesive properties due to the inclusion of the polymer Carbopol 934P (C934P) in the formulation to deliver the anti-inflammatory and anti-pain drug naproxen [152].

#### 4.3.2. pH-Responsive Polymers

pH-responsive polymers represent a group of stimuli-responsive polymers that have in their chemical structure weak acidic or basic groups that either accept or release protons in response to a change in the environmental pH. They show different architectures, having linear (as in homopolymers, di, ter, multiblock copolymers, and organic/inorganic hybrid polymers), nonlinear (branched or hyperbranched polymers), and lightly or highly cross-linked network polymers.

These polymers respond to changes in the pH of the environment undergoing structural changes, e.g., in chain conformation alteration and configuration, and with property changes, such as surface activity and solubility [153,154]. The protonation or deprotonation of functional groups they experience may result in chain flocculation and/or collapse. These molecules can also self-assemble in larger structures or swell [154], as illustrated in Figure 7.

The most commonly used pH-sensitive polymers are linear homopolymers amphiphilic in nature and double hydrophilic block copolymers which form micelles, vesicles, stars, branched and hyperbranched polymers, polymer brushes, dendrimers, nanogels, micro-gels, and hydrogels (macrogels) [154,155].

There are many studies in the literature that report the use of this type of material for drug delivery to the oral cavity. Hu et al. (2019) synthesized a series of amphiphilic pH-sensitive block copolymer poly(methyl methacrylate-co-methacrylic acid)-b-poly(2-amino ethyl methacrylate) [P(MMA-co-MAA)-b-PAEMA] via activators regenerated by electron transfer atom transfer radical polymerization (ARGET ATRP) and further self-assembled into pH-responsive cationic polymeric micelles (PCPMs) for oral insulin delivery [156]. Zamani et al. (2019) carried out an in vivo study of the use of poly (ethylene glycol)-poly(caprolactone)-modified folic acid nanocarriers as a pH-responsive system for tumor-targeted co-delivery of tamoxifen and quercetin [157]. Arjama et al. (2018) developed a sericin/rice bran albumin embedded gellan gum-based smart nanosystem for pH-responsive delivery of doxorubicin [158]. Sherje and Londhe (2017) developed and evaluated a pH-responsive cyclodextrin-based in situ gel of paliperidone for intranasal delivery [159]. Ishak et al. (2020) developed pH-responsive gamma-irradiated poly(acrylic acid)-cellulose-nanocrystal-reinforced hydrogels and concluded that they show a good response toward pH, thus suggesting their potential as a drug-delivery system [160]. Jamshidzadeh et al. (2020) investigated the modification of halloysite nanotubes by chitosan (CTS) and pectin (PCN) to produce new pH-sensitive bionanocomposites via a layer-by-layer method. The main objective of this study was to improve loading efficiency and control the release of phenytoin sodium (PHT) [161].

Although many of the above-mentioned works do not necessarily focus on drug delivery to oral mucosa lesions, pH-responsive polymers have a great potential to be further explored in this field.

#### 4.3.3. Bioresponsive Polymers

Bioresponsive polymers are those that change their properties due to stimuli provided by biologically relevant molecules such as glucose, ATP, enzymes, antibodies, etc. These stimuli can cause changes in molecular interactions of the polymer, which eventually translate into macroscopic responses, such as solution-to-gel transitions, polymer swelling, or collapse [162,163].

Ulijn et al. (2007) categorized bioresponsive hydrogels based on three types of stimuli [164]. The first includes hydrogels modified with small biomolecules able to selectively bind to protein receptors, antibodies, or other biomacromolecules. This interaction can trigger a macroscopic transition in the hydrogel. The second type refers to hydrogels modified with sensitive enzymes. In this case, molecular recognition can cause a chemical change that involves breaking the bonds of a substrate molecule. Hydrogels having biomacromolecules incorporated into their structure comprise the third case. The biomacromolecules, such as enzymes, can recognize small biomolecules as substrates and convert them into molecules with different physical properties. This modification can trigger swelling or collapse of the hydrogel, and the signal can be utilized, e.g., to produce a biosensor.

Maitz et al. (2013) developed bio-responsive polymer hydrogels capable to homeostatically regulate blood coagulation [165]. The authors synthesized unmodified amino-terminated starPEG or the peptide-functionalized starPEG (PEG-FXRS), which were used to form non-responsive or thrombin-responsive hydrogels, respectively, useful in case of bleeding lesions of the oral cavity. The reported system was shown to be a simple yet powerful and clinically relevant example, illustrating the enormous potential of reciprocally triggered bio-responsive polymers.

## 5. Methods for the Production of Drug Delivery Systems

The manufacture of biomaterials for the oral cavity may be carried out by various methods, some as simple as mixing the excipients and the active agents (e.g., in the case of a number of gels and ointments), and some more elaborated, such as solvent casting, electrospinning, hot-melt extrusion, and 3D printing, among many others. The most commonly used methods will be discussed below.

### 5.1. Solvent Casting Method

The solvent casting method is the most common technique applied for the preparation of single or multilayer films on a lab scale as well as in continuous industrial production processes [18,166,167]. The solvent casting method is based on three steps: preparing a homogeneous mixture of components, obtaining a dry laminate by evaporating the solvent, and, lastly, cutting the laminate into the desired shape and dimensions [167].

In the first step, the drug and excipients (stabilizers, plasticizers, and other products necessary for adequate film formation) are dissolved or dispersed in an appropriate solvent or solution using, for instance, a stirring tank at an industrial level or a beaker in a laboratory/pharmacy setting. Then, the formed mass is transferred to a cast and dried to produce a film with constant thickness and uniformity of drug content.

The solvent casting method can be used industrially for pharmaceutical applications, in which the drug is suspended or dissolved in a solution of polymers in a volatile solvent. The film is deposited on a continuous-release roller impregnated on a plastic base and then passed through a drying apparatus to remove the solvents (Figure 8). The dry film is then cut to sizes suitable for proper use [167].

Figure 8 and Figure 9 illustrate the steps for producing a polymeric film using the solvent casting technique on laboratory and industrial scales, respectively.

### 5.2. Electrospinning

Electrospinning is a versatile technique useful to produce ultrathin fibers. This technique involves an electrohydrodynamic process, during which a liquid droplet is electrified, stretched, and elongated to generate fibers, as a result of exposure to a high-voltage electric field. The solvent in the fibers evaporates, and the fibers are deposited onto a collector [169].

The electrospinning equipment is basically composed of four elements: (1) a positive displacement pump (or a syringe pump at laboratorial scale), to ensure constant flow of the polymeric solution; (2) a high voltage power supply, which is responsible for yielding the driving force necessary to attract the fibers from the polymeric solution to the collector; (3) a metal tip or needle that directs the solution into the high voltage electric field; and (4) a grounded collector used in static or rotating mode, over which the fibers are deposited and form a non-woven film [169,170].

To produce the fibers, a high voltage (normally between 15 and 20 kV) is applied to the chamber, between the tip of the needle and the collector. The polymeric solution is pumped and tends to assume a conical shape on the tip of the needle (forming a Taylor cone), where the liquid voltage is in equilibrium with the electric field. When the electric field intensity exceeds the surface tension, the polymer jet is ejected, and the solution is electrospun and deposited on the collector. When jetting occurs, evaporation of the solvent also occurs, followed by weaving of the fibers, which produces a mat with high surface area to volume ratio. The basic setup for electrospinning is shown in Figure 10.

Several factors affect the characteristics of the fibers formed, e.g., electrospinning conditions, solution formulation, and environmental parameters. Electrospinning parameters comprise the intensity of the electric field, the separation distance between the needle and the collector, solution flow rate, and needle diameter. The solution-related parameters are polymer properties and concentration, viscosity, solvent type, and solution conductivity. The environmental parameters include relative humidity and temperature. All these parameters must be taken into account so that the required design of the material can be achieved [172].

Similar to the solvent casting method, the electrospinning technique is also widely investigated for the production of biomaterials that can be used for oral drug delivery. Table 4 illustrates some of the works found in the literature in this sense.

Although widely used on laboratory scale, electrospinning has a few limitations regarding its scale-up for the production of biomaterials by the pharmaceutical industry, such as the low productivity and the use of organic solvents, in which case the manufacturer must guarantee that the final product does not contain harmful levels of residual solvent. Furthermore, strategies to increase productivity have been studied, such as multi-jet electrospinning, multi-needle electrospinning, and free-surface electrospinning, in which the needle is replaced by an open surface. Although these technologies also have drawbacks, studies have been carried out to address these issues and some industrial scale equipment is already available [173].

### 5.3. Hot Melt Extrusion

The hot melt extrusion (HME) technique makes use of heat and pressure in mixing a set of materials using a screw in a barrel, which transports the molten material through a matrix to mold it into the desired shape [174]. The pharmaceutical process involves pumping polymeric materials with a rotating screw at temperatures above their glass transition temperature (T_g_) or above the melting temperature (T_m_) to promote their effective mixing with the active compounds at a molecular level. This molecular mixing process converts the components into an amorphous product with uniform shape and density, increasing the dissolution profile of poorly water-soluble drugs [175].

During the design and optimization of an HME process, two main classes of factors should be evaluated, one related to the equipment design itself and its operating conditions, and the other referring to the characteristics of the chemical compounds used in the production of the biomaterial. Process variables include the selection of the appropriate equipment, mass feeding rate, process temperature, and tolerable shear stress, while physicochemical factors must also be considered, such as drug and excipient properties, possible interactions among the components of the formulation, mixture physical state, and stability.

The design and optimization of an HME process can be described in terms of pre-formulation, formulation, and post-formulation phases and should be investigated. Factors, such as the chemical and thermal stability of extrudates, the solid physical state of extrudates, drug–polymer interaction, miscibility or solubility of the drug–polymer system, rheological properties of extrudates, physicomechanical properties of films produced by hot melt extrusion, and drug particle dissolution from extrudates [176], are frequently analyzed.

This technology, which follows basically the same procedures and steps at small and large scale, may offer advantages over conventional pharmaceutical manufacturing processes, such as shorter and more efficient time to achieve the final product, environmental advantages due to the elimination of solvent use, increased solubility and bioavailability of the drugs, taste masking, and production of amorphous materials. Thus, HME has emerged as an alternative platform technology to other traditional techniques for manufacturing pharmaceutical dosage forms, such as tablets, capsules, films, and implant, for drug delivery not only via oral routes, but also for transdermal and transmucosal administration [175,176,177]. Figure 11 shows a schematic of hot-melt extrusion technique used to produce films, tablets, capsules, and pellets.

### 5.4. 3D Printing Method

Three-dimensional (3D) printing is an additive manufacturing (AM) technique for fabricating a wide range of structures with simple or complex geometries from three-dimensional model data. The process consists of printing successive layers one above the other until the desired shape and dimensions are obtained [177]. According to the ASTM F2792-12A [179], the manufacturing process techniques of AM are divided into seven main categories, as shown in Table 3 [180,181].

With the 3D printing technique, it is possible to produce highly customized, functional parts of varied composition. These materials can be metals, ceramics, polymers, or even their combinations. To produce polymeric biomaterials, extrusion techniques and those based on resin and powder processes are commonly used. Each type of process allows the manufacture of pieces in different ways, using exclusive steps [181,182]. In Figure 12, three different 3D printing techniques are illustrated.

Fused deposition modeling (FDM) is an extrusion method used to deposit filaments of thermoplastics. The layout for FDM consists of a printhead able to move along X and Y directions above a built platform. The polymer is extruded through the heated nozzle and laid down as filaments according to the CAD design. The build platform is then moved along Z direction, and layers can be built until the object has the desired shape completed [184].

By the stereolithography (SLA) printing process, it is possible to fabricate 3D objects using light to selectively solidify a liquid resin through a photopolymerization reaction, achieving a resolution of about 20–40 µm [185,186]. This technique can be used to directly produce biomaterials from photopolymers and photoinitiators, as well as to create a negative replica of the desired structure. In the latter, the structure can then be filled with ceramic or metallic slurries, followed by sintering. In this way, a broader range of materials may be used [186].

In direct laser writing, ultraviolet light is directed towards a vat of photosensitive resin to form solid layers with a moving build platform. Therefore, this technique is based on the non-linear absorption of photons by the photopolymers, for which a laser beam is focused on the volume of a transparent material, leading to the absorption of two or more photons and polymerizing locally. This laser can be moved according to a path previously established in a computer, being able to reproduce a CAD model, for instance [187].

Products designed for efficient and controlled oral drug delivery obtained by this technique are exemplified in Table 4.

## 6. Market Perspectives of Drug Delivery Biomaterials for the Oral Cavity

Several companies that provide market research services monitor the market and global tendencies of investment in the areas of drug delivery systems, materials for drug delivery in the oral cavity, and mucoadhesive drug delivery systems, e.g., Databridge [207], Future Marketing Insights [208], Grand View Research [209], Transparency Market Research [210], and Biospace [211], among others.

The data provided by these companies, despite being quite divergent in many points, show that these fields of investment are very attractive, given their market value and growth projections for the next years.

The Future Marketing Insights company, for instance, estimates the market value of oral controlled release drug delivery in US$ 34.1 billion for 2022, projecting a market value of US$ 68.4 billion by 2032. However, these data are focused on oral systems designed to deliver drugs in the gastrointestinal tract [208].

In the case of evaluation of oral transmucosal drugs market, most of the analyses cover different routes of penetration, such as buccal and sublingual mucosa, gingival and palatal tissues, as well as the evaluation of preferred distribution channels (e.g., retail, hospital and online pharmacies). In 2020, according to Transparency Market Research, the buccal mucosa segment dominated the global market and retail pharmacies were the favorite distribution channel (around 50% of the total), with significant growth being expected in the online pharmacy distribution channel segment [210].

The company Biospace pointed that the global oral transmucosal drugs market was of US$ 14 billion in 2020 [211]. Transparency Market Research predicts an increase to US$ 27 billion in 2031 for the same market, with a compound annual growth rate (CAGR) of around 6% [211]. Other companies estimate CAGR values for oral transmucosal drug market in the same neighborhoods: 6% (Transparent Market and Biospace) [210,211]; 7.2% (Future Marketing Insights) [208]; 7.6% (Databridge Company) [207], 9.2% (Grand View Research) [205].

There are some key factors that are driving the market growth, including the increasing prevalence of chronic disorders, growing importance to develop drug delivery systems tailored to the patient needs, and the increasing R&D expenditure related to drug delivery systems by pharma companies. The use of drug delivery systems, moreover, reduces the chances of side effects by optimizing the drug presence at the target site. This helps in minimizing the dose and undesired effects. In addition, this helps in reducing the overall product price by decreasing the amount of active pharmaceutical ingredient required.

According to Grand View Research [209], North America dominated the buccal drug delivery systems market with a share of 31.6% in 2020. Delivery devices in the oral cavity have gained prominence due to constant research and development in this field. The delivery of drugs via the oral mucosa is considered a convenient and highly appreciated choice, especially among elderly patients. With this delivery pathway, it is possible to eliminate first-pass metabolism, allowing pH-sensitive and easily degradable drugs to be administered more effectively. This route also allows rapid administration, making it a field of many opportunities.

According to Future Market Insights, the key companies profiled in the oral controlled release drug delivery technology segment include AbbVie (North Chicago, IL, USA), Amgen (Thousand Oaks, CA, USA), AstraZeneca Plc. (Cambridge, UK), Bayer (Leverkusen, Germany), BioNTech (Mainz, Germany), Boehringer Ingelheim (Ingelheim am Rhein, Germany), Bristol-Myers Squibb Company (New York, NY, USA), Eli Lilly (Indianapolis, IN, USA), F. Hoffmann-La Roche Ltd. (Basel, Switzerland), Gilead Sciences (Foster City, CA, USA), GlaxoSmithKline Plc. (London, UK), Johnson & Johnson (New Brunswick, NJ, USA), Merck & Co. Inc. (Rahway, NJ, USA), Moderna (Cambridge, MA, USA), Novartis AG (Cambridge, MA, USA), Pfizer Inc. (New York, NY, USA), Sanofi S.A. (Paris, France), Sun Pharmaceuticals (Mumbai, India), Takeda (Tokyo, Japan), and Viatris (Canonsburg, PA, USA) [208].

Specifically in the oral transmucosal drugs market, the company Transparency Market Research pointed as main players, in addition to AstraZeneca (Cambridge, UK), Bristol-Myers Squibb Company (New York, NY, USA) and GlaxoSmithKline Plc (Brentford, UK), Access Pharmaceutical Inc. (Tokyo, Japan), Aquestive Therapeutics Inc. (Warren, NJ, USA), C.L Pharm (Seoul, Republic of Korea), Cure Pharmaceutical, Eisai Co. Ltd. Inc. (Tokyo, Japan), Solvay S.A. (Brussels, Belgium), IntelGenx Corp. (Montreal, QC, Canada), Izun Pharmaceuticals, Ltd. (New York, NY, USA), Pfizer Inc. (New York, NY, USA), LTS Lohmann Therapie-Systeme AG (Andernach, Germany), Mylan N.V. (Canonsburg, PA, USA), NAL Pharma (Hong Kong, China), Otsuka Pharmaceutical Co. (Tokyo, Japan), Seoul Pharmaceuticals (Seoul, Republic of Korea), Soligenix (Princeton, NJ, USA), Teva Pharmaceutical Industries Ltd. (Tel Aviv, Israel), and ZIM Laboratories Limited (Nagpur, India) [210].

The high number of companies operating in this market reinforces the perception of a trend towards increasing investments and technology development in this sector.

## 7. Conclusions

Although numerous formulations and biomaterials for drug delivery are described in the literature, drug delivery in the oral cavity is still relatively unexplored, especially when compared to drug delivery to the skin. There is a therapeutic need for drug delivery in this complex region which still presents many challenges for treating local oral diseases. This approach is also interesting because it is easy to be used by children, the elderly, and other people with difficulties, e.g., swallowing medication. In the literature, several studies describe the use of fast dissolving oral films, mucoadhesive formulations, such as tablets, films, and gels; and microneedles, which is a more recent category with a wide range of applications and is highly promising. Biomaterials for application in the oral cavity can be composed of synthetic and natural polymers. Natural polymers represent a highly promising class, as they are more biocompatible than synthetic ones. Finally, these devices can be produced by solvent casting, electrospinning, hot-melt extrusion, and 3D printing, among other techniques. The 3D printing technique is one of the most applied due to its versatility and for being customizable. While many challenges are still present in this field, there are several opportunities to produce oral cavity systems to promote drug delivery, which is well illustrated by the high number of companies focusing on this particular area.

## Figures and Tables

**Figure 1 pharmaceutics-15-00012-f001:**
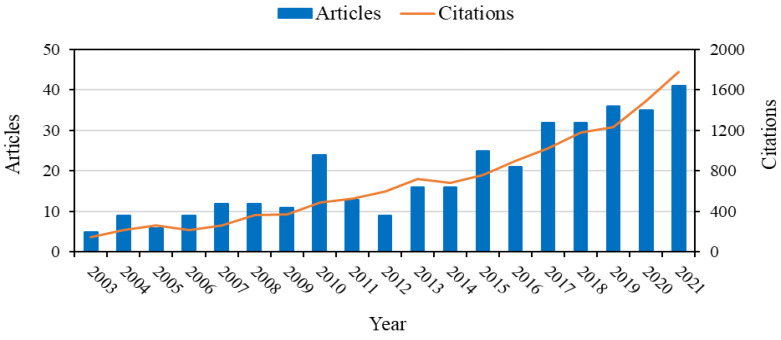
Number of articles in the last years addressing the evaluation of polymers used to produce formulations devices for oral cavity application and the citations associated with these publications. The literature search was based on Web of Science, with the keywords “Polymers + oral cavity”.

**Figure 2 pharmaceutics-15-00012-f002:**
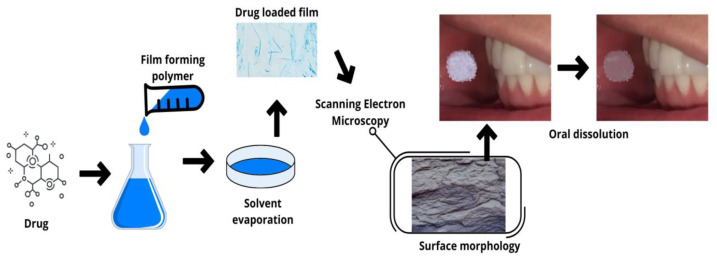
Schematic representation of film production by the casting technique until its application in the mouth, and its dissolution.

**Figure 3 pharmaceutics-15-00012-f003:**
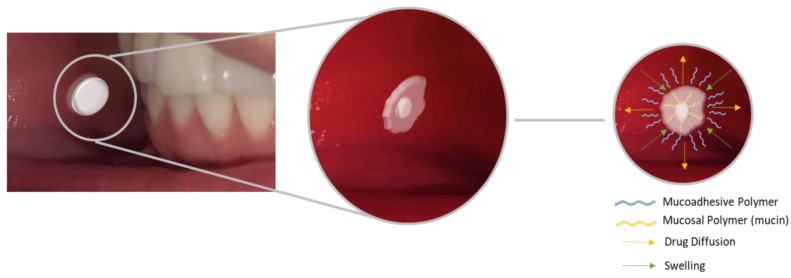
Factors that influence the behavior of mucoadhesive tablets and their drug release profile.

**Figure 4 pharmaceutics-15-00012-f004:**
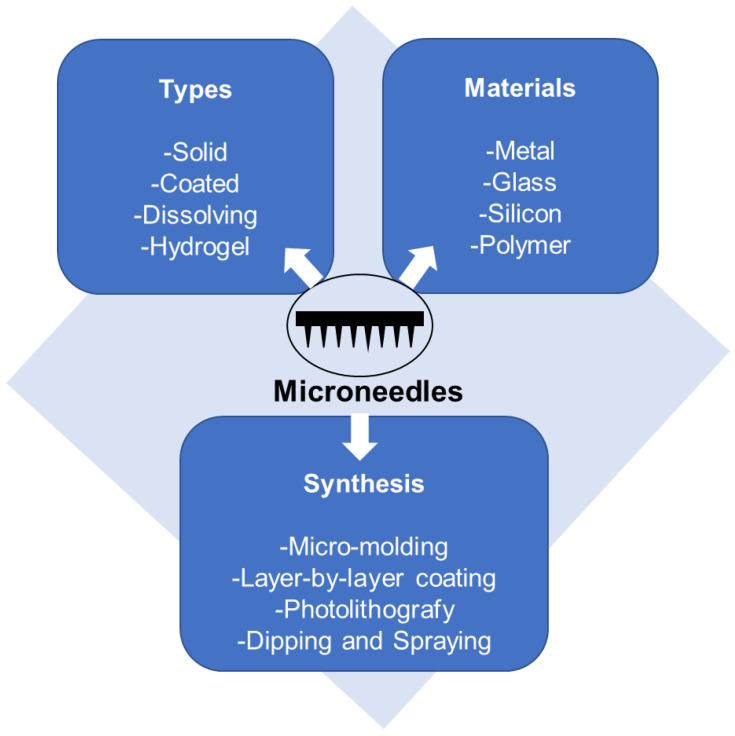
Diagram summarizing microneedle types, materials and production methods.

**Figure 5 pharmaceutics-15-00012-f005:**
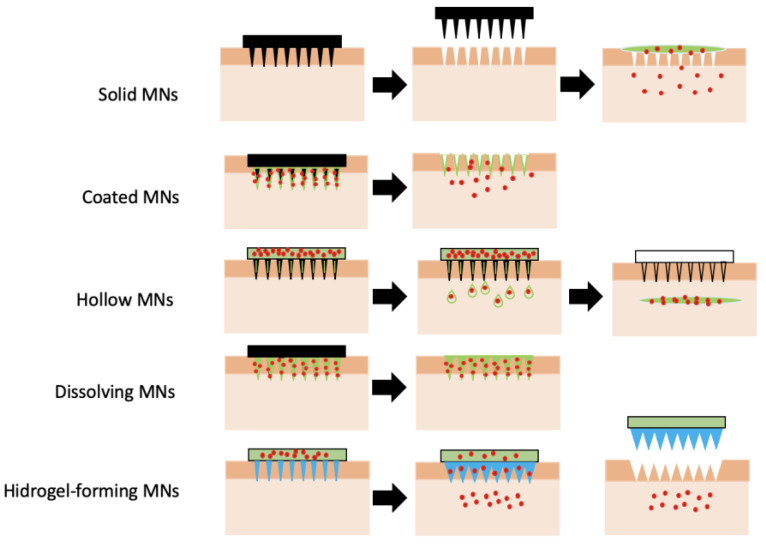
Schematic diagram showing the different types of microneedles and the mechanisms through which the different microneedles are used for drug delivery (adapted from [85]).

**Figure 6 pharmaceutics-15-00012-f006:**
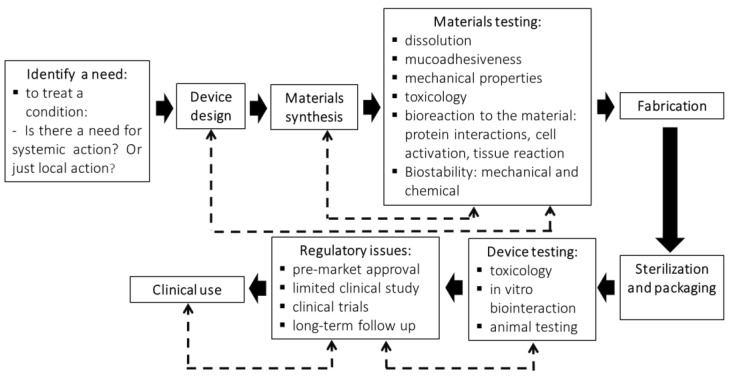
Illustrative diagram of the steps for the development of a new biomaterial designed for drug delivery to the oral cavity. Dashed lines refer to steps in which retro-feedback may occur with the purpose of improving the biomaterial originally designed (adapted from [127,128]).

**Figure 7 pharmaceutics-15-00012-f007:**
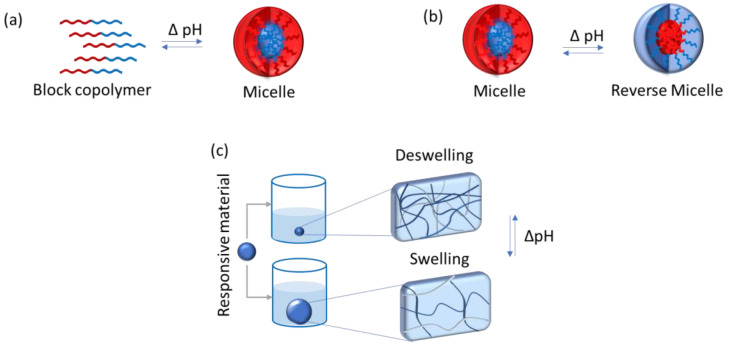
Possible transformations caused by pH modification in the surroundings of the polymeric environment.

**Figure 8 pharmaceutics-15-00012-f008:**
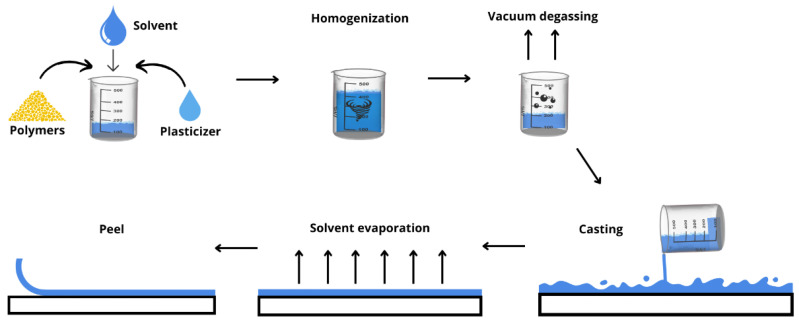
Scheme of the production, in a small scale, of a polymeric film using the solvent casting technique.

**Figure 9 pharmaceutics-15-00012-f009:**
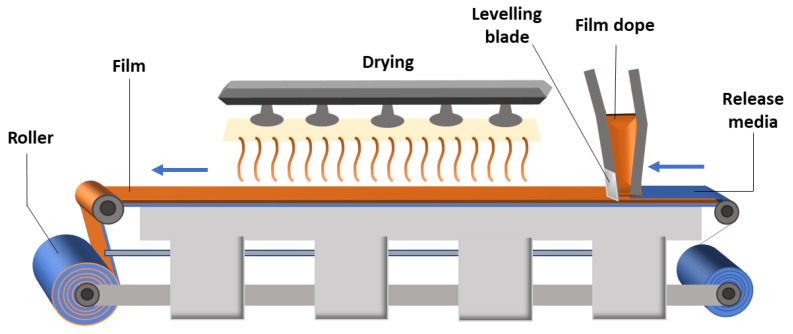
Scheme of industrial production of a polymeric film using the solvent casting technique (adapted from [168]).

**Figure 10 pharmaceutics-15-00012-f010:**
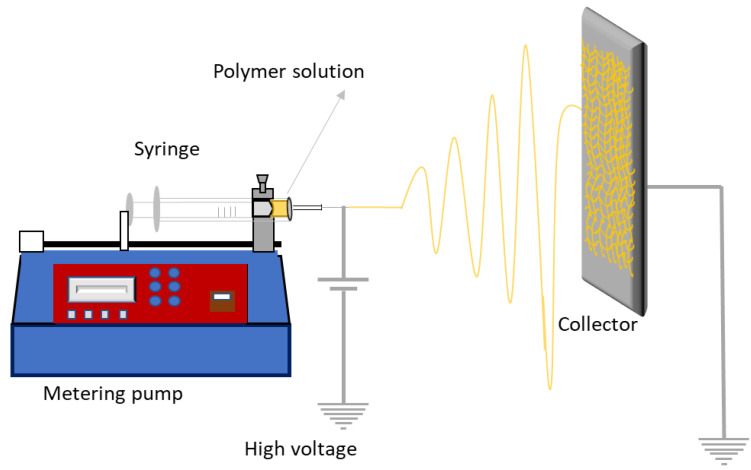
Basic configuration of electrospinning equipment (adapted from [171]).

**Figure 11 pharmaceutics-15-00012-f011:**
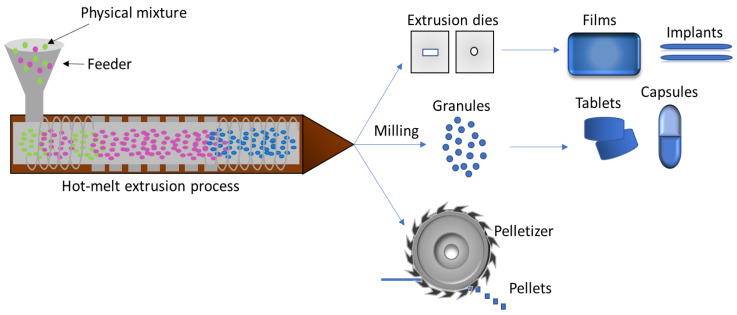
Schematic of hot-melt extrusion of films, tablets, capsules, and pellets (adapted from [178]).

**Figure 12 pharmaceutics-15-00012-f012:**
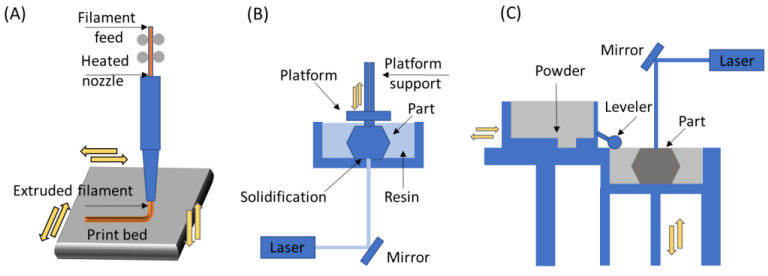
Representation of different 3D printing techniques: (**A**) fused deposition modeling, (**B**) stereolithography, and (**C**) selective laser sintering (adapted from [183]).

**Table 1 pharmaceutics-15-00012-t001:** Examples of drugs and polymers used in the formulation of oral fast-dissolving films *.

Polymer	Drug	Therapeutic Use	Tested Model	References
HPMC/PVA	Levocetirizine dihydrochloride	Antihistamine	In vitro dissolution tests, and in vivo studies in rats	[24]
HPMC/PVA	Telmisartan	Hypertension	In vitro dissolution test	[25]
PVP	Paracetamol/caffeine	Analgesic and antipyretic	In vitro dissolution test	[26]
Chitosan	Metformin	Diabetes	In vitro dissolution test	[27]
PEG/400	Lercanidipine	Hypertension, and angina pectoris	In vitro Dissolution test and ex vivo drug permeation through porcine buccal mucosa	[28]
HPMC	Mirtazapine	Depression	In vitro dissolution test	[29]
HPMC/Alginate	Lidocaine	Anesthetic	In vitro dissolution test	[30]
Pullullan	Salbutamol sulfate	Asthma	In vitro dissolution tests, and in vivo studies on humans	[31]
Gelatin/Starch	Vitamin C	Assists in numerous functions	In vitro dissolution tests, and in vivo studies on humans	[32]
Chitosan/Pullullan	Aspirin	Minor aches, pains, and fever	In vitro dissolution test	[33]
Gelatin/gelatinized tapioca starch	Lidocaine	Anesthetic	In vitro dissolution test and ex vivo drug permeation through chick chorioallantoic membrane (CAM)	[34]
HPMC	Venlafaxine	Depression	In vitro dissolution test	[35]
Pectin/CMC	Paroxetin	Depression and anxiety	In vitro dissolution test and ex vivo drug permeation through chicken buccal pouch	[36]

* CMC (carboxymethyl cellulose), HPMC (hydroxypropyl methylcellulose), PEG (polyethylene glycol), PVP (polyvinyl pyrrolidone), PVA (polyvinyl acrylate). As observed in Table 1, numerous polymers can be used to produce films with rapid degradation, and cellulose-based biomaterials stand out for this purpose. Table 1 also presents a large spectrum of drugs that can be administered through the oral cavity, minimizing the need for other invasive and often uncomfortable routes.

**Table 2 pharmaceutics-15-00012-t002:** Main natural polymers used to manufacture oral drug delivery biomaterials and examples of their applications.

Natural Polymer	Main Characteristic	Examples of Use	Tested Model	References
Alginate	Ability to form reversible hydrogels through interaction of carboxylic acid functional groups with metal cations	Film for the delivery of cetirizine dihydrochloride	In vitro release test	[134]
Cellulose	Tunable mechanical properties	Biomaterials for buccal delivery of non-steroidal anti-inflammatory drugs	In vitro release test	[135]
Chitosan	Cationic nature, a property that allows the formation of electrostatic complexes with negatively charged polymers	Gel for the delivery of the antimicrobial *Schinus molle* L essential oil	No models have been tested	[136]
Collagen	Excellent biocompatibility, flexibility and ability to absorb body fluids for delivery of nutrients	Buccal patch for the delivery of lorazepan	In vitro release test and ex vivo drug permeation through bovine buccal mucosa	[137]
Gellan Gum	Excellent gelation capability	Film for the delivery of triamcinolone acetonide	In vitro release test	[138]
Guar Gum	Excellent ability to hydrate rapidly, generating highly viscous solutions	Film-nanoparticle composite for the delivery of alpha-casozepine	In vitro release test	[57]
Gelatin	Thermoresponsiveness	Film for the delivery of propranolol hydrochloride	In vitro and in silico release test	[48]
Hyaluronic acid	Biocompatibility, hydrophilicity, low immunogenicity and excellent viscoelasticity	Mucoadhesive microneedles for the delivery of lidocaine	In vivo drug permeation through rat buccal mucosa	[104]
Pectin	Excellent biodegradability, biocompatibility and possibility of ionic crosslinking	Film for the delivery of triamcinolone acetonide	In vitro release test	[139]

**Table 3 pharmaceutics-15-00012-t003:** ASTM F2792-12A standard terminology for additive manufacturing processes.

Type of Process	Basic Description
Material extrusion	Material is extruded through a nozzle or orifice and deposited on the surface
Material jetting	Drops of material are deposited on the surface until the desired layers are formed
Binder jetting	A liquid bonding agent is deposited to join powder materials
Sheet lamination	Material sheets are deposited to form the final desired object
Vat photopolymerization	Liquid photopolymer in a vat is selectively cured by light-activated polymerization
Powder bed fusion	Thermal energy selectively fuses regions of a powder bed
Directed energy deposition	Focused thermal energy is used to fuse materials by melting as the material is deposited

**Table 4 pharmaceutics-15-00012-t004:** Examples of techniques to prepare biomaterials, application, matrix composition, active agents incorporated and main conclusions reached.

Biomaterial Production Technique	Biomaterial Application	Matrix Composition	Active Agent	Main Conclusion	Reference
Solvent casting	Oral fast-dissolving film	HPMC	Risperidone	Satisfactory physicochemical properties and in vitro behavior	[188]
	Oral fast-dissolving film	HPMC	Diazepam	Good mechanical strength, drug release, disintegration time and stability	[189]
	Oral fast-dissolving films	HPMC	Loratadine	Good physico chemical properties. The solvent casting method can be adopted for the preparation of films	[190]
	Mucoadhesive buccal films	EC/HPMC	Ornidazole/ dexamethasone	Desirable physical characteristics and mucoadhesive properties	[191]
	Microneedles	PVP	POXA1b laccase enzyme	The microneedles were able to control the release kinetics of the compound incorporated	[77]
Electrospinning	Oral film	Chitosan and PEO	Insulin	Fiber morphology, film mechanical properties, and in vitro stability dependent on PEO feed ratio. Lower PEO content formulations produced smaller diameter fibers with significantly faster insulin release kinetics	[192]
	Mucoadhesive buccal film	PVP, Eudragit RS100 and PEO (mucoadhesive layer) and PCL	Lidocaine	Analysis of ex vivo diffusion through porcine buccal mucosa suggested that lidocaine permeated the oral mucosa, enabling its use to reduce pain in the oral cavity.	[193]
	Oral fast-dissolving films	PVA	Caffeine and riboflavin	Burst released of both drugs (caffeine to an extent of 100% and riboflavin to an extent of 40% within 60 s) from PVA nanofibrous matrices	[194]
	Oral fast-dissolving films	Chitosan/ Pullulan	Aspirin	Fast film dissolution and efficient aspirin encapsulation indicated potential use for oral mucosal drug release	[33]
	Oral fast-dissolving films	PVP	Escitalopram and quetiapine	The drug-loaded fibers exhibited a disintegration time of 2 s, which accelerated the release of both drugs (50% after 5 min) making it an attractive formulation for oral mucosal delivery	[195]
	Fast-dissolving drug delivery system	Jelly fig polysaccharide/Pullulan	Hydrophobic drugs	Formulation consisting of a promising carrier to encapsulate hydrophobic drugs for fast-dissolving/disintegrating delivery applications.	[196]
Hot melt extrusion	Oral tablets	EVA	Metropolol tartrate	Drug release dependent on drug loading and extrusion temperature	[197]
	Oral fast dissolving film	Lycoat^®^ RS 780 (modified starch)	Chlorpheniramine maleate	Films showed immediate disintegration and dissolution, due to the presence hydrophilic excipients. The formulation showed to be a good option to produce solvent-free thin films	[198]
	Mucoadhesive buccal film	HPC/HPMC/PEG	Salbutamol sulphate	Evidence provided to support the selection of formulation compositions to produce hot-melt extruded mucoadhesive films	[199]
	Mucoadhesive buccal film	PEO N10/HPMC/Eudragit RL100	Domperidone	HME is a viable technique for the preparation of buccal-adhesive films with improved drug bioavailability	[200]
	Mucoadhesive oral tablet	PEO/HPMC	Pioglitazone/felodipine	The optimized formulation showed adequate in vitro drug release, ex vivo permeation, and bioadhesive properties	[201]
3D printing	Oral film	Pullulan/HPMC	Caffeine	Effective spatial deposition control of films and successful determination of orientation to maximize the mechanical properties of the hybrid films obtained through 3D printing	[202]
	Microneedles	Alginate and hydroxyapatite	Glucose -responsive insulin	Microneedles exhibited sufficient mechanical strength to penetrate the skin of mice and responsively released insulin according to the glucose levels both in glucose solution and in type 1 diabetic mice	[203]
	Mucoadhesive oral film	HPMC	Catechin hydrate	Flexible application of 3D bioprinters (semi-solid extrusion-type 3D printers) to prepare film formulations	[204]
	Oral disintegrating tablets	PVP/Starch/ Microcrystalline Cellulose	Warfarin sodium	Tablets prepared by the 3D technique showed uniform drug content, good mechanical properties, and presented fast disintegration and fast dissolution	[205]
	Oral fast-dissolving films	PEO/PVP/ Poloxamer (P 407 and P188)	Olanzapine	The films showed increased dissolution rates of the poorly water-soluble drug, consisting in a suitable formulation for fast drug absorption	[206]

## Data Availability

Not applicable.

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
