# Peer review of "Polymeric Biomaterials for Topical Drug Delivery in the Oral Cavity: Advances on Devices and Manufacturing Technologies"

_pharmaceutics, 2022, doi:10.3390/pharmaceutics15010012_

Round 1

Reviewer 1 Report

General comments

The review entitled “Advances and Technologies of Polymeric Biomaterials for Topical Drug Delivery in the Oral Cavity” illustrates different aspects of the use of polymers as biomaterials to design and manufacture drug delivery systems for topical administration to the oral cavity.

The manuscript is well organized, easy and interesting to read, while providing relevant references on the topic.

Specific comments.

 At line 143, the authors claim “mucoadhesive buccal tablets can be loaded with a greater amount of drug than films and gels” while at line 177 the authors state that films “ensure a higher drug dosage when compared to gels and other formulations such as tablets, pastes, and sprays,”.

These statements are conflicting. Please explain.

The sections Author contribution, Funding, Data Availability Statement, Acknowledgement, and Conflict of interest should be filled in.

References should be formatted according to the Journal style.

English should be revised.

Author Response

Reviewer #1: 

1) The review entitled “Advances and Technologies of Polymeric Biomaterials for Topical Drug Delivery in the Oral Cavity” illustrates different aspects of the use of polymers as biomaterials to design and manufacture drug delivery systems for topical administration to the oral cavity. The manuscript is well organized, easy and interesting to read, while providing relevant references on the topic.

Answer: The authors are grateful for the analysis and compliments. 

2) At line 143, the authors claim “mucoadhesive buccal tablets can be loaded with a greater amount of drug than films and gels” while at line 177 the authors state that films “ensure a higher drug dosage when compared to gels and other formulations such as tablets, pastes, and sprays,”. These statements are conflicting. Please explain.

Answer: In fact, tablets are the devices with the greatest potential for loading a large amount of drugs. This contradiction has been corrected in the text. Thank you for pointing it.

3) The sections Author contribution, Funding, Data Availability Statement, Acknowledgement, and Conflict of interest should be filled in.

Answer:  These topics have been added to the text.

4) References should be formatted according to the Journal style.

Answer:  Corrections on formatting were made in the text.

5) English should be revised.

Answer: The text has been carefully proofread, thank you for the suggestion.

Reviewer 2 Report

In the manuscript of Remiro et al., the polymeric biomaterials and their implementation in topical cavity drug delivery systems were considered. The recent advances in the topical pharmaceutical forms were presented as well as their potential to the market. The structure of the review article has to be modified. The review starts with the introduction that contains very basic info about the oral cavity drug delivery. The differences between oral cavity and gastrointestinal administration route are briefly mentioned. To my opinion section 2 and 3 should switch order. In the section 3 the different pharmaceutical forms/devices were presented. They are called biomaterials, that is little confusing. The polymers are part of their structure, but they are pharmaceutical forms or devices. Many parts were just like reading the textbook with the general info. There is significant advance in this field, and to my opinion it was not presented in this review. Or the structure of the review hinders its potential.

Author Response

Reviewer #2: 

1) In the manuscript of Remiro et al., the polymeric biomaterials and their implementation in topical cavity drug delivery systems were considered. The recent advances in the topical pharmaceutical forms were presented as well as their potential to the market. 

Answer: Thank you for the analysis.

2) The structure of the review article has to be modified: 

  1. a) The review starts with the introduction that contains very basic info about the oral cavity drug delivery. The differences between oral cavity and gastrointestinal administration route are briefly mentioned. 

Answer: Thank you for the analysis. We believe that an introduction on basic fundamentals of drug delivery in the oral cavity is important in the review paper, since this route administration is not yet as explored in the literature as other more traditional routes, such as drug ingestion and injection routes. 

3) To my opinion section 2 and 3 should switch order. 

Answer: We thank you for the suggestion. After reading it, we considered that indeed the flow of information could be improved in the manuscript, but instead of inverting sections 2 and 3, we inverted sections 3 and 4. In this sense, the description of the polymers used to produce biomaterials for use in the oral cavity was directly linked to Figure 7 of the previous version of the manuscript, which was the illustrative diagram of the steps for the development of a new biomaterial designed for the delivery of a drug to the oral cavity. Therefore, in the present version of the manuscript, the choice of polymers may be considered as one of the first steps of the production of the biomaterials to be used for drug delivery in the oral cavity in association with choosing the methods for the production of drug delivery systems themselves. We hope that this alteration improved the chain of concepts focused by the work.

4) In the section 3 the different pharmaceutical forms/devices were presented. They are called biomaterials, that is little confusing. The polymers are part of their structure, but they are pharmaceutical forms or devices. 

Answer: This point is indeed confusing because the term biomaterial is frequently used in the literature to describe materials of biological origin, but in the field of materials developed particularly for the solution of health problems, biomaterials are defined as  “a nonviable material used in a medical device, intended to interact with biological systems” (Willians, 1987), as described in the book Biomaterials Science: An Introduction to Materials in Medicine, (please see in reference Ratner et al, 2013 described below).

Williams, D. F. (1987). Definitions in Biomaterials: Proceedings of a Consensus Conference of the European Society for Biomaterials, Chester, England, March 3–5, 1986. Amsterdam; NY: Elsevier.

Ratner et al. Biomaterials Science: An Introduction to Materials in Medicine, 3rd edition, Elsevier, p.23, 2013.

5) Many parts were just like reading the textbook with the general info. There is significant  advance in this field, and to my opinion it was not presented in this review. Or the structure of the review hinders its potential.

Answer: Thank you for your comment. We do present some basic information on our review paper, however, since the field of research on transbuccal drug delivery systems is very multidisciplinary, we believe that our paper could reach people with a variety of backgrounds, and the basic knowledge of these readers may also be very different. Therefore, in our opinion, we could contribute significantly by combining some fundamental information with discussions on the current development of devices used for this route of drug administration and on the polymers and  technologies used  to manufacture them. With that purpose in mind, the examples listed in the paper are mostly new/current developments. Of all cited articles, about 80% were published in the last 10 years, 55% in the last 5 years and 23% in the last 2 years. These data corroborate the idea of ​​the relevance of this topic in recent years. Nevertheless, if the reviewer could kindly provide us with additional examples of significant advances in this field that could improve our manuscript, we would gladly consider these suggestions and make the appropriate adjustments.

Reviewer 3 Report

The authors present us a review, entitled "Advances and Technologies of Polymeric Biomaterials for Topical Drug Delivery in the Oral Cavity", a well-made, documented, and organized article on a high interest topic consulting an impressive list of very recent bibliographical references (no less than 206). 

The importance of finding alternatives for the administration of drugs with reduced bioavailability is brought to our attention and, in this regard, emphasis has been placed on the transbuccal delivery, an already known but insufficiently explored route of administration.

The authors have done previous research on transdermal administration as evidenced by some references and now the interest is on biomaterials used in the transbuccal route, focusing on smart polymers for the controlled delivery of drugs.

The description of the main biopolymers used in the pharmaceutical industry and their preparation methods is very useful and is done in a didactic but easy way, accompanied by illustrative figures and tables containing a series of polymers with the related active substances including a wide range of therapeutic classes.

The oral cavity for topical drug delivery can be an alternative for certain pathologies such as oral carcinoma where different agents, nanoparticle sized, can be successfully used. We agree with the authors regarding the potential of this route of administration and that there is still room for further research having the advantage that it is easy to use by children, the elderly, and others with difficulties, for example, swallowing medication.

To conclude, the manuscript under consideration meets the necessary requirements for publication.

Author Response

Reviewer #3: 

The authors present us a review, entitled "Advances and Technologies of Polymeric Biomaterials for Topical Drug Delivery in the Oral Cavity", a well-made, documented, and organized article on a high interest topic consulting an impressive list of very recent bibliographical references (no less than 206). 

The importance of finding alternatives for the administration of drugs with reduced bioavailability is brought to our attention and, in this regard, emphasis has been placed on the transbuccal delivery, an already known but insufficiently explored route of administration.

The authors have done previous research on transdermal administration as evidenced by some references and now the interest is on biomaterials used in the transbuccal route, focusing on smart polymers for the controlled delivery of drugs.

The description of the main biopolymers used in the pharmaceutical industry and their preparation methods is very useful and is done in a didactic but easy way, accompanied by illustrative figures and tables containing a series of polymers with the related active substances including a wide range of therapeutic classes.

The oral cavity for topical drug delivery can be an alternative for certain pathologies such as oral carcinoma where different agents, nanoparticle sized, can be successfully used. We agree with the authors regarding the potential of this route of administration and that there is still room for further research having the advantage that it is easy to use by children, the elderly, and others with difficulties, for example, swallowing medication.

To conclude, the manuscript under consideration meets the necessary requirements for publication.

Answer: The authors are very grateful for the detailed analysis and for so many positive considerations.

Reviewer 4 Report

Dear Authors,

The manuscript entitled "Advances and Technologies of Polymeric Biomaterials for Topical Drug Delivery in the Oral Cavity” is an interesting topic. Nowadays, drug delivery systems in the oral cavity it is an interesting and pertinent theme.

This manuscript helps us to understand the new developments, technologies, and market perspective.

To improve the manuscript, I present general comments and suggestions.

1. Title

- Focus the title more on the purpose of the Review

- Line 21 – remove “T”

2. Keywords

- include “biomaterials”

3. Introduction

- Line 81 - Figure 1. Number of articles…

- blue bar – articles only.

- Table 1 - improve the table, reporting other polymers and active molecules, for example, caffeine.

- I suggest reading and referencing some of these articles:

Batista P, Castro P, Madureira AR, et al: Development and characterization of chitosan microparticles-in-films for buccal delivery of bioactive peptides. Pharmaceuticals 12:32, 2019.

2.  Batista P, Castro PM, Madureira AR, et al: Recent insights in the use of nanocarriers for the oral delivery of bioactive proteins and peptides. Peptides 101:112-123, 2018.

3.  Batista P, Castro PM, Madureira AR, et al: Preparation, Characterization and Evaluation of Guar Films Impregnated with Relaxing Peptide Loaded into Chitosan Microparticles. Applied Sciences 11:9849, 2021.

4.  Batista P, Oliveira-Silva P, Heym N, et al: Neuropsychophysiological measurements as a tool for neuromodulator oral films evaluation. INTERNATIONAL JOURNAL OF PHARMACEUTICAL RESEARCH, 2020.

5.  Batista P, Rodrigues PM, Ferreira M, et al: Validation of Psychophysiological Measures for Caffeine Oral Films Characterization by Machine Learning Approaches. Bioengineering 9:114, 2022.

6.  Castro PM, Baptista P, Madureira AR, et al: Combination of PLGA nanoparticles with mucoadhesive guar-gum films for buccal delivery of antihypertensive peptide. International journal of pharmaceutics 547:593-601, 2018.

7.  Castro PM, Baptista P, Zuccheri G, et al: Film-nanoparticle composite for enhanced oral delivery of alpha-casozepine. Colloids and Surfaces B: Biointerfaces 181:149-157, 2019.

- Line 149 – Figure3. Factors that influence…

- Line 209 - reduce the less relevant and older references.

For example: for skin, use only one reference, the most relevant one. The focus is on the oral cavity.

- summarize the information about Microneedles, and include polymeric Microneedles in the same chapter.

- Line 367 and Table 2 – include “guar gum”

- Line 556 – “… for drug delivery to the…”

- Figures 8 and 9 were originally produced or taken/adapted from other articles?

- The information from lines 595 to 609 has no referencing…

- The author's contribution to this study should be highlighted. 

- There are many references and old ones, I suggest reducing the number of references, removing the old ones, and adopting the more current and relevant ones.

It is an article that not only reviews the literature on the subject but also addresses the market, which gives it originality and relevance.

Author Response

Reviewer #4: 

The manuscript entitled "Advances and Technologies of Polymeric Biomaterials for Topical Drug Delivery in the Oral Cavity” is an interesting topic. Nowadays, drug delivery systems in the oral cavity is an interesting and pertinent theme. This manuscript helps us to understand the new developments, technologies, and market perspective.  To improve the manuscript, I present general comments and suggestions.

1) Title: Focus the title more on the purpose of the Review.

Answer: Thank you very much for this suggestion. We changed the title to:  “Polymeric Biomaterials for Topical Drug Delivery in the Oral Cavity: Advances on Devices and Manufacturing Technologies”

2)  Line 21: remove “T”

Answer: The suggestion was accepted and implemented in the current version of the manuscript.

3) Keywords: include “biomaterials”

Answer: The suggestion was accepted and implemented in the current version of the manuscript.

4) Line 81 - Figure 1. Number of articles…

Answer: The suggestion was accepted and implemented in the current version of the manuscript.

5) Blue bar – articles only

Answer: The suggestion was accepted and implemented in the current version of the manuscript.

6) Table 1 - improve the table, reporting other polymers and active molecules, for example, caffeine.

Answer: The paper presents representative examples of various types of synthetic and natural polymers, drugs, and plant extracts, among others. The  references cited in this table only make up 6% of the citations of the entire paper. Thus, although there are other interesting papers in the literature, we consider that a sufficient number of works was added and other additions might turn the work a little repetitive in content. .

7) I suggest reading and referencing some of these articles:

  1. Batista P, Castro P, Madureira AR, et al: Development and characterization of chitosan microparticles-in-films for buccal delivery of bioactive peptides. Pharmaceuticals 12:32, 2019.
  2. Batista P, Castro PM, Madureira AR, et al: Recent insights in the use of nanocarriers for the oral delivery of bioactive proteins and peptides. Peptides 101:112-123, 2018.
  3. Batista P, Castro PM, Madureira AR, et al: Preparation, Characterization and Evaluation of Guar Films Impregnated with Relaxing Peptide Loaded into Chitosan Microparticles. Applied Sciences 11:9849, 2021.
  4. Batista P, Oliveira-Silva P, Heym N, et al: Neuropsychophysiological measurements as a tool for neuromodulator oral films evaluation. INTERNATIONAL JOURNAL OF PHARMACEUTICAL RESEARCH, 2020.
  5. Batista P, Rodrigues PM, Ferreira M, et al: Validation of Psychophysiological Measures for Caffeine Oral Films Characterization by Machine Learning Approaches. Bioengineering 9:114, 2022.
  6. Castro PM, Baptista P, Madureira AR, et al: Combination of PLGA nanoparticles with mucoadhesive guar-gum films for buccal delivery of antihypertensive peptide. International journal of pharmaceutics 547:593-601, 2018.
  7. Castro PM, Baptista P, Zuccheri G, et al: Film-nanoparticle composite for enhanced oral delivery of alpha-casozepine. Colloids and Surfaces B: Biointerfaces 181:149-157, 2019.

Answer: Papers 1 and 7 have been added as references to the manuscript, as they more closely focus the topics covered in the text. Thank you for the suggestions.

8) Line 149 – Figure 3. Factors that influence…

Answer: The suggestion was accepted and implemented in the current version of the manuscript. 

9)  Line 209 - reduce the less relevant and older references. For example: for skin, use only one reference, the most relevant one. The focus is on the oral cavity.

Answer: We believe that it is important to point to relevant works published in the past, as they helped to define current developments.  

10) Summarize the information about Microneedles, and include polymeric Microneedles in the same chapter.

Answer: Thank you for your suggestion. The authors agree that the information on microneedles is very extensive. In our opinion, and based on current literature, these are among the most recently developed and promising devices for drug delivery to the oral cavity. They are already widely used for transdermal drug delivery and we believe that the development of microneedles for mucosa will  grow significantly in the next few years. These considerations are highlighted in the present version of the manuscript. Therefore we think that the manuscript can better contribute to the scientific community by maintaining this large volume of information on microneedles. 

11) Line 367 and Table 2 – include “guar gum”

Answer: The suggestion was accepted, and  guar gum was added to Table 2 in  the current version of the manuscript.

12)  Line 556 – “… for drug delivery to the…”

Answer: The suggestion was accepted and implemented in the current version of the manuscript. 

13) Figures 8 and 9 were originally produced or taken/adapted from other articles?

Answer: The images were adapted from previously published images and credits were given to the authors when appropriate

14) The information from lines 595 to 609 has no referencing…The author's contribution to this study should be highlighted. 

Answer: References have been added to the text. Thank you for the suggestion.

15) There are many references and old ones, I suggest reducing the number of references, removing the old ones, and adopting the more current and relevant ones.

Answer: Indeed there are many  references in the paper to works published not as recently, however, many current references were cited as well. We consider the relationship between new and old articles consistent, since many of the recently published works were based or inspired by papers published in the past. In our opinion, the contribution of selected works that were published formerly should be recognized in certain contexts. Moreover, of all cited articles, about 80% were published in the last 10 years, 55% in the last 5 years and 23% in the last 2 years, and we consider this distribution appropriate

16) It is an article that not only reviews the literature on the subject but also addresses the market, which gives it originality and relevance.

Answer: The authors are grateful for these considerations.

Round 2

Reviewer 2 Report

After revision, the authors improved the manuscript and answered to all raised questions. Therefore I suggest the acceptance of the manuscript.